# Micronutrient Deficiencies in Heart Failure and Relationship with Exocrine Pancreatic Insufficiency

**DOI:** 10.3390/nu17010056

**Published:** 2024-12-27

**Authors:** Marlene A. T. Vijver, Nils Bomer, Robert C. Verdonk, Peter van der Meer, Dirk J. van Veldhuisen, Olivier C. Dams

**Affiliations:** 1Department of Cardiology, University Medical Centre Groningen, University of Groningen, 9713 GZ Groningen, The Netherlands; m.a.t.vijver@umcg.nl (M.A.T.V.); n.bomer@umcg.nl (N.B.); p.van.der.meer@umcg.nl (P.v.d.M.); o.c.dams@umcg.nl (O.C.D.); 2Department of Gastroenterology and Hepatology, St. Antonius Hospital, 3435 CM Nieuwegein, The Netherlands; r.verdonk@antoniusziekenhuis.nl

**Keywords:** heart failure, micronutrients, exocrine pancreatic insufficiency, vitamin A, vitamin D, vitamin E, selenium, zinc

## Abstract

Background: Micronutrient deficiencies are common and play a significant role in the prognosis of many chronic diseases, including heart failure (HF), but their prevalence in HF is not well known. As studies have traditionally focused on causes originating within the intestines, exocrine pancreatic insufficiency (EPI) has been overlooked as a potential contributor. The exocrine pancreas enables the absorption of various (fat-soluble) micronutrients and may be insufficient in HF. We hypothesize that EPI contributes to micronutrient deficiencies in HF. Objectives: To evaluate micronutrient concentrations in HF cases and their association with clinical characteristics and EPI. Materials and Methods: Plasma samples from 59 consecutive hospitalized patients with HF were analyzed for vitamins A, D, and E and the minerals selenium and zinc. EPI was defined as fecal elastase 1 level < 206 μg/g. Results: The mean age of patients was 59 ± 14 years, with 24 (41%) being women, and a median NT-proBNP concentration of 3726 [2104–6704] pg/mL was noted. Vitamin A deficiency occurred in eight (14%) of the patients, and 12 (20%) exceeded the upper limit. More than half (51%) were vitamin D-deficient. No patients showed vitamin E deficiency, but 14 (24%) had elevated levels. Selenium deficiency was common, affecting 36 (61%) patients, while zinc was below the normal range in seven patients (12%). Micronutrient levels did not differ significantly based on the presence of EPI. Conclusions: This study provides novel insights into the micronutrient status of patients with HF. Deficiencies in vitamins A and D, selenium, and zinc are prevalent in HF, but these findings are not associated with exocrine pancreatic function.

## 1. Introduction

Micronutrients play a role in maintaining normal human physiology, and deficiencies of these essential nutrients may accelerate the progression of various (chronic) diseases, such as heart failure (HF) [1,2,3]. Despite this, data on micronutrient status in patients with HF are limited and mostly derived from small-scale studies, but showing that deficiencies are related to disease severity [4]. The prevalence and mechanisms underlying the development of specific deficiencies in HF remain poorly understood.

Proposed mechanisms of micronutrient deficiencies in HF include poor dietary habits, hormonal and/or catabolic imbalances, increased excretion, and malabsorption [3,5]. The occurrence of malabsorption in HF can be partially explained by the ‘gut hypothesis’, which encompasses processes in which bowel wall edema and impaired intestinal barrier function lead to bacterial translocation and heightened inflammation [6]. While most studies have concentrated on causes originating within the intestines, exocrine pancreatic insufficiency (EPI) has been overlooked as a potential contributor to malabsorption in HF.

Recently, there has been a growing interest in the interplay between the heart and the exocrine pancreas [7,8,9,10,11]. We previously demonstrated that EPI occurs in a significant proportion of patients with HF [8]. In patients with pancreatic damage due to chronic pancreatitis, EPI results in fat malabsorption and diminished nutritional status, driving micronutrient deficiencies [12]. However, the association between pancreatic dysfunction and micronutrient status in HF has yet to be evaluated.

Understanding this relationship could help clarify the mechanisms underlying micronutrient deficiencies in patients with HF. This study aims to evaluate micronutrient status in patients with HF, focusing on fat-soluble vitamins (A, D, and E) as well as the minerals selenium and zinc, since these micronutrients depend on the exocrine pancreas for absorption [12]. Additionally, we aim to evaluate the association between micronutrients and clinical characteristics and exocrine pancreatic function in patients with HF.

## 2. Materials and Methods

### 2.1. Study Population

Data were obtained from a cross-sectional, single-center study, conducted at the University Medical Centre Groningen (UMCG) and registered under NL-OMON50432 [8]. The study included 59 consecutive, well-characterized patients hospitalized with HF at the department of cardiology. Inclusion criteria included age ≥ 18 years and fulfilling the European Society of Cardiology criteria for HF [13]. Patients who were a priori at risk for pancreatic dysfunction, such as those with a history of pancreatic disease and/or gastrointestinal malabsorption, as well as those experiencing diarrhea at the time of enrolment were excluded. For these patients’ medical history, clinical characteristics were retrieved from the patients’ medical files, as well as available echocardiographic data and laboratory measurements (including kidney and liver function and NT-proBNP). As this study concerns a secondary analysis of a study on the prevalence of EPI in HF, no formal sample size calculation was performed [8].

All patients provided written informed consent. Ethical approval was obtained by the Institutional Review Board of the UMCG (METc 2021/529), and the work complies with the Declaration of Helsinki.

### 2.2. Micronutrient Assessment

Plasma samples were analyzed to assess levels of vitamins A (retinol), D (25-OH-vit D3, calcidiol), and E (α-tocopherol); selenium; and zinc, collected using an EDTA tube. Samples were processed within 1 h of collection to prevent hemolysis. Afterwards, samples were frozen to −20 °C before further assessment. Vitamin A, D, and E concentrations were quantified using a validated liquid chromatography with tandem mass spectrometry (LC-MS/MS) method with lower limits of quantification of 0.01 μmol/L, 1.13 nmol/L, and 0.03 μmol/L, respectively. Selenium and zinc concentrations were measured using a validated inductively coupled plasma mass spectrometry (ICP-MS) method with lower limits of quantitation of 20 μg/L and 0.38 μmol/L plasma, respectively. The 78Se and 64Zn isotope concentrations were measured using a Varian 820-MS ICP mass spectrometer. Reference values are shown in Figure 1 and are in line with UMCG laboratory standards [14,15,16,17].

### 2.3. Determination of Exocrine Pancreatic Function

Fecal elastase 1 (FE-1) was measured to evaluate exocrine pancreatic function, following standard UMCG laboratory procedures [18]. The Pancreatic Elastase ELISA kit from BIOSERV Diagnostics was used (Eu Registration no.: DE/CA/81/2009-17). While variability exists in the criteria used, international guidelines suggest that an FE-1 level < 200 μg/g supports the diagnosis of EPI [19]. Previous investigations have demonstrated a coefficient of variation of approximately 3–4% in relation to the BIOSERV kit, a finding corroborated by the UMCG laboratory [20]. Therefore, EPI in this study was defined as an FE-1 level < 206 μg/g, allowing for a coefficient of variation of 3%.

### 2.4. Statistical Analyses

The normality of the data was checked by visual assessment of Q-Q plots. Data that adhere to a normal distribution are presented as means (±standard deviation), non-normally distributed data are presented as medians [interquartile range], and categorical data are presented as frequencies (percentages). Differences between two groups containing normally distributed continuous data were tested using Student’s *t*-test, whereas non-normally distributed data were tested using the Mann–Whitney test. Differences in categorical variables were tested using the Fisher–Freeman–Halton test. For ordinal data, the Kruskal–Wallis exact test was executed. Linear regression was conducted to analyze associations between micronutrients and clinical variables of interest. Non-normally distributed parameters were transformed into a logarithmic scale. To variables with values between 0 and 1, a constant of 1 was added to ensure all transformed values were suitable for transformation. A two-sided p-value of less than 0.05 was considered significant in all analyses. All data were analyzed using SPSS IBM version 28.

## 3. Results

Overall, the 59 patients with HF included in this study had a mean age of 59 ± 14 years, and 24 patients (41%) were women (Table 1). The median N-terminal pro-B-type natriuretic peptide (NT-proBNP) level was 3726 [2104–6704] pg/mL, and the mean left ventricular ejection fraction was 29 ± 13%. EPI according to FE-1 level was identified in five (9%) patients. Patients with EPI were younger (44 ± 13 versus 61 ± 13 years; *p* = 0.007), had a lower body mass index (BMI) (21.8 ± 3.3 versus 26.9 ± 4.5 kg/m^2^; *p* = 0.016), and were more likely to have congenital heart disease (60% versus 15%; *p* = 0.04).

Plasma micronutrient levels and corresponding reference values are depicted in Table 1 and Figure 1. Two-thirds of the patients had adequate vitamin A levels, while 8 (14%) were deficient and 12 (20%) showed a level that exceeded the upper limit. More than half of the cohort was vitamin D-deficient, with a median level of 47.2 [26.7–68.6] nmol/L. Vitamin D was taken as a daily supplement by eight patients (median plasma value was 65.5 [56.1–81.2] nmol/L). No patients were deficient for vitamin E, with 14 (24%) patients showing elevated plasma levels (>35 µmol/L). More than half (56%, 33 patients) exhibited selenium deficiency (<70 µg/L). Zinc was below the normal range in 7 (12%) and above in 2 (3%) patients.

Associations between plasma levels of vitamins A, D, and E; selenium; and zinc and the predefined variables of interest were evaluated using regression analysis (Appendix A respectively). Vitamin D levels were negatively associated with weight and BMI (*p* = 0.03 and *p* = 0.01, respectively), whilst vitamin E levels were positively associated with age (*p* = 0.02).

Micronutrient levels were compared between patients with HF with and without EPI, revealing no significant differences (Table 1).

The micronutrient status in patients affected by EPI did not show any specific significant trends (Table 2). One patient had vitamin A deficiency, one patient was suffering from a vitamin D deficit, and another showed zinc deficiency. Three out of five patients were selenium-deficient.

## 4. Discussion

This is the first study to present the distribution of five key micronutrients—vitamins A, D, and E as well as the minerals selenium and zinc—among 59 patients with HF with and without EPI. Although no significant associations were found between micronutrient status and the presence of EPI, patients with EPI tended to have lower plasma selenium levels.

### 4.1. Vitamin A

In the present study, vitamin A levels were mostly within the normal range. Only one other study evaluated vitamin A status in 21 patients with chronic HF, reporting elevated serum levels compared to a control group (3.1 ± 3.25 versus 2.14 ± 0.65 μmol/L), which were above normal range and higher than the vitamin A levels in our cohort (1.9 [1.4–2.6] μmol/L) [4]. Data on serum vitamin A concentrations in developed countries are scarce, and deficiency is rare (ranging from 0 to 7% in the United Kingdom) [21]. Similarly, the prevalence of vitamin A deficiency in patients with EPI is around 14%, aligning with our findings, although we did not find a direct association with EPI in our small cohort [22].

While deficiency is rare, an excess of vitamin A typically only arises from overconsumption [23]. However, since the patients with high levels of vitamin A did not take additional vitamins, this could not be explained by excess intake. It is hypothesized that higher vitamin A levels in patients with HF may result from an increased demand for nutritional antioxidants, as vitamin A’s antioxidative properties may mitigate cardiovascular damage by neutralizing reactive oxygen species [24]. Studies investigating the association between vitamin A and cardiovascular disease have presented conflicting results, showing both inverse and positive associations [25]. While earlier trials showed that supplementation with vitamin A could reduce cardiovascular events, more recent research failed to confirm these benefits [26,27]. Notably, the vitamin A levels of participants in these studies were not measured, and data regarding the effects of vitamin A supplementation in HF patients remain lacking. Further research is needed to clarify the role of vitamin A supplementation in patients with HF, particularly in light of the conflicting evidence and the lack of data on serum levels in previous studies.

### 4.2. Vitamin D

More than half of the study population suffered from vitamin D deficiency (<50 nmol/L; according to Dutch guidelines), with none showing elevated levels. Vitamin D deficiency is common, with rates in the Netherlands ranging from 35 to 60%, depending on the season [28]. Generally, vitamin D deficiency is more prevalent in patients with HF compared to those without [29]. Among patients with EPI, the prevalence of vitamin D deficiency is approximately 70%, but we observed no relevant relationship between the presence of EPI and vitamin D deficiency [22].

Vitamin D deficiency increases the risk of cardiovascular events, including stroke, myocardial infarction, and HF [30]. Moreover, in patients with established HF, low levels of vitamin D are associated with a poorer prognosis [31]. In accordance with these findings, epidemiological and preclinical studies suggested that vitamin D has cardioprotective properties [32]. However, despite these correlations and hypothetical properties, recent randomized controlled trials have failed to provide evidence for any beneficial impact on cardiovascular health, including prevention and/or the clinical course of HF [29,33]. This contradiction highlights the need for further research to clarify the role of vitamin D in cardiovascular disease.

### 4.3. Vitamin E

In our study, mean vitamin E level was 31.5 ± 9.4 μmol/L, which is comparable to the results of another study in 21 patients with chronic HF (36.1 ± 13.3 μmol/L) [4]. Vitamin E deficiency is uncommon due to its wide availability in various foods, with prevalence rates in most European countries ranging from 0 to 1%, and it did not occur in the present study [21]. Although some reports indicate that vitamin E deficiency occurs in 32% of patients affected with EPI, this was not present in the patients with HF and EPI in our study [22]. Interestingly, nearly a quarter of the population showed elevated vitamin E levels, unrelated to excess intake.

There is a delicate balance between normal values and excess vitamin E. In patients with HF, the amount of reactive oxygen species is increased, necessitating antioxidative agents, such as vitamin E, as a compensatory scavenging mechanism [24]. It is postulated that increased serum concentrations of vitamin E reflect a higher demand for antioxidative nutrients [24,34]. Moreover, diuretic effects of vitamin E have been described, which may explain the higher amount of circulating vitamin E as a compensatory response in congestive HF [35]. These mechanisms may account for the absence of vitamin E deficiencies despite the presence of EPI and even explain the values above the upper limit of normality. The latter could have also contributed to the failure of clinical trials to show beneficial effects of preventive administration of vitamin E on major adverse cardiovascular events [36]. Notably, the vitamin levels of the participants were not measured prior to, during, or after the completion of the trials.

### 4.4. Selenium

Selenium deficiency was present in a substantial proportion of the investigated patients and was the only micronutrient that showed a trend towards lower levels in patients with EPI compared to those without, although this difference was not significant. Evidence on the clinical importance of selenium in patients with HF is accumulating rapidly [37].

A recent retrospective study analyzed selenium levels in a large sample of patients with new-onset or worsening HF [38]. The investigators found that selenium-deficient subjects had impaired mitochondrial function and a worse prognosis, indicating an important role of selenium in metabolism and cellular health. Specifically, patients with lower selenium levels exhibited high levels of reactive oxygen species. The prevalence of selenium deficiency (<70 μg/L) in chronic HF was observed to be around 25%, whereas rates in acute HF cases can reach up to 84% [39]. In patients with EPI in the context of chronic pancreatitis or pancreaticoduodenectomy [40], selenium deficiency was reported in 15% and 56%, respectively [41]. Nevertheless, the underlying mechanism of the role of exocrine secretions in (intestinal) selenium handling is not well understood.

No substantial randomized trial of selenium supplementation in HF has been conducted to date. A large meta-analysis showed that the risk of cardiovascular mortality was decreased in patients who were administered antioxidants which included selenium [42]. Furthermore, only patients with selenium deficiency appeared to benefit from supplementation [42,43]. Another meta-analysis found a moderate association between serum selenium concentration and risk of coronary heart disease [44]. Further research is required to clarify the potential role of selenium in the management of HF.

### 4.5. Zinc

The World Health Organization states that zinc deficiency is common in many regions of the world [45]. However, in European countries, zinc deficiency is relatively uncommon, with prevalence rates ranging from 0 to 7% [21]. In the present study, seven (12%) patients were zinc-deficient, which is comparable to observations in acute HF settings [39]. Although data appear to be conflicting, patients with HF tend to have lower zinc levels [2], and this could be related to HF medication that influences the renin–angiotensin–aldosterone system [5,46], which increases hyperzincuria through urinary acidification and metabolic alkalosis. The prevalence of zinc deficiency in patients with pancreatic dysfunction remains unclear [41].

One study showed that patients with HF and decreased serum zinc levels (<9.5 μmol/L) have higher mortality rates [47]. Moreover, there is increasing evidence indicating that in patients with zinc deficiency, especially in combination with selenium deficiency, deficiency-related cardiomyopathy can develop, characterized by a decline in myocardial antioxidant reserves and oxidative cell damage [5]. Lastly, an Italian study identified patients with HF who had undergone bariatric surgery, accompanied by malabsorption of selenium and zinc [48]. Following selenium/zinc infusion, they witnessed an improvement in cardiomyocytes with a decrease in pathological changes. Although large randomized trials are lacking, these results could provide a rationale for monitoring zinc levels and addressing deficiencies in HF.

### 4.6. Exocrine Pancreatic Insufficiency and Heart Failure

Both EPI and HF share common risk factors, such as chronic alcohol use, diabetes mellitus, smoking, and malnutrition. These conditions can predispose individuals to both pancreatic and cardiac dysfunction [11]. Malnutrition, which is prevalent in HF due to reduced appetite, poor absorption, and increased metabolic demands, can intensify the consequences of EPI. Conversely, EPI leads to malabsorption of fats, proteins, and fat-soluble vitamins, worsening nutritional status and potentially further complicating HF management. In this cohort, we provide evidence for malnutrition, indicated by lower BMI, in HF patients with coexisting EPI. However, an association between EPI and micronutrient deficiencies was not identified. This can be explained by the small sample (n = 5) of patients who were diagnosed with EPI, limiting our analyses. Additionally, compensatory mechanisms, such as increased mobilization of fat-soluble vitamins from adipose reserves, can allow for adequate micronutrient status despite the presence of EPI. Interestingly, in a population of patients with EPI due to chronic pancreatitis, EPI was associated with an increased risk of cardiovascular events. More specifically, the risk was associated with a decrease in nutritional markers [49]. Future research on micronutrient status in patients with HF and cardiac cachexia, where reserves are limited, is warranted.

### 4.7. Limitations

Firstly, it was conducted in a tertiary center, which may result in a population that is not fully representative of the general HF population. Secondly, we identified only five patients with EPI who were significantly more malnourished, limiting the ability to interpret possible associations between nutrient deficiencies. Thirdly, EPI was diagnosed solely based on the measurement of FE-1 levels; no direct, invasive measurements were performed. While this approach is in concordance with current guidelines and provides acceptable sensitivity for diagnosing moderate to severe disease, mild exocrine pancreatic dysfunction may go undetected. Fourthly, dietary intake was not analyzed, which could have provided more insight into the underlying etiologies of nutrient deficiencies. Lastly, several micronutrients may be influenced by pre-analytical factors, such as acute phase response, protein binding, and time of sampling, which were not controlled for in this study. This may contribute to variability in the measured concentrations.

## 5. Conclusions

This study highlights the prevalence of micronutrient deficiencies in patients with HF. While certain micronutrient deficiencies, such as vitamin D and selenium, were observed alongside the potential for malnutrition, no definitive associations with EPI were established. Larger studies are warranted to further elucidate the prognostic impact of micronutrient deficiencies and possible associations with EPI and malabsorption in patients with HF.

## Figures and Tables

**Figure 1 nutrients-17-00056-f001:**
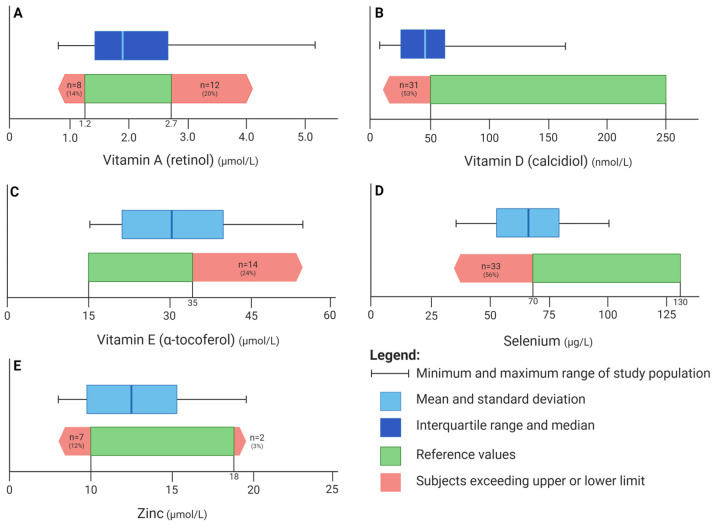
Visual representation of distribution of nutrients in patients with HF compared to population-based reference values. Distribution of nutrients is indicated by the upper blue boxplot, displaying the median and interquartile ranges for vitamins A and D and the means with standard deviations for vitamin E, selenium, and zinc. The black lines refer to the minimum and maximum values. Reference values (0.025–0.975 of the population range) are indicated by the green bar. The red arrow bars indicate the number of patients who exceeded the lower or upper limit of normality. (**A**) Distribution of vitamin A. (**B**) Distribution of vitamin D. (**C**) Distribution of vitamin E. (**D**) Distribution of selenium. (**E**) Distribution of zinc.

**Table 1 nutrients-17-00056-t001:** Baseline characteristics of the study population and differences between patients with and without exocrine pancreatic insufficiency.

	Total (n = 59)	With EPI (n = 5)	Without EPI (n = 54)	*p-*Value
Demographics				
Age, years	59 ± 14	44 ± 13	61 ± 13	0.007
Female sex, n	24 (41)	3 (60)	21 (39)	0.39
Race				1.00
Caucasian, n	57 (97)	5 (100)	52 (96)	
Other, n	2 (3)	0 (0)	2 (4)	
Weight, kg	81.1 ± 17.1	61.1 ± 13.5	83.0 ± 16.4	0.006
BMI, kg/m^2^	26.4 ± 4.6	21.8 ± 3.3	26.9 ± 4.5	0.016
Clinical features of HF				
Etiology, n				0.03
Ischemic heart disease	14 (24)	2 (40)	12 (22)	0.58
Dilated cardiomyopathy	21 (36)	0 (0)	21 (39)	0.15
Congenital heart disease	11 (19)	3 (60)	8 (15)	0.04
Other	13 (22)	0 (0)	13 (24)	0.58
Duration of HF, years	10 [4.0–18.0]	27.0 [6.2–38.0]	8.5 [4.0–18.0]	0.18
NYHA functional class, n				0.23
I	0 (0)	0 (0)	0 (0)	
II	10 (17)	1 (20)	9 (17)	
III	34 (58)	1 (20)	33 (61)	
IV	15 (25)	3 (60)	12 (22)	
Medical history, n				
Myocardial infarction	16 (27)	2 (40)	14 (26)	0.61
Hypertension	15 (25)	1 (20)	14 (26)	1.00
Diabetes mellitus	10 (15)	1 (20)	9 (17)	1.00
Type I	1 (2)	1 (20)	0 (0)	
Type II	8 (14)	0 (0)	8 (15)	
Stroke	6 (10)	0 (0)	6 (11)	1.00
Pulmonary hypertension	7 (12)	2 (40)	5 (9)	0.10
Vitamin suppletion, n				
Vitamin D	5 (12)	1 (20)	6 (11)	0.48
Multivitamins	1 (2)	0 (0)	1 (2)	1.00
Echocardiography				
LVEF, %	29 ± 13	28 ± 21	29 ± 13	0.96
LVEDD, mm	59 ± 12	62 ± 17	59 ± 12	0.57
TAPSE, mm	15.0 [12.6–17.0]	14.5 [12.6–15.7]	15.0 [12.6–17.4]	0.59
RV S’ cm/s	8.8 ± 2.7	8.0 ± 2.1	8.8 ± 2.7	0.54
Laboratory				
Vitamin A, µmol/L	1.9 [1.4–2.6]	1.8 [1.2–2.3]	2.0 [1.4–2.6]	0.35
Vitamin A-deficient, n	8 (14)	1 (20)	7 (13)	1.00
Vitamin D3, nmol/L	47.2 [26.7–68.6]	62.1 [48.9–82.0]	45.2 [24.2–68.7]	0.23
Vitamin D3-deficient, n	31 (53)	1 (20)	30 (58)	0.17
Vitamin E, µmol/L	31.5 ± 9.4	29.2 ± 9.1	31.7 ± 9.5	0.58
Vitamin E-deficient, n	0 (0)	0 (0)	0 (0)	-
Selenium, µg/L	68.9 ± 14.0	62.4 ± 12.4	69.5 ± 14.0	0.28
Selenium-deficient, n	36 (61)	3 (60)	30 (58)	1.00
Zinc, µmol/L	12.6 ± 2.8	11.6 ± 2.5	12.7 ± 2.8	0.42
Zinc-deficient, n	7 (12)	1 (20)	6 (86)	1.00
CRP, mg/L	10 [4–20]	8 [5–77]	10 [4–20]	0.93
eGFR, L/min/1.73 m^2^	57 ± 24	76 ± 21	55 ± 24	0.06
Creatinine, µmol/L	129 ± 73	92 ± 23	132 ± 75	0.24
Sodium, mmol/L	137 ± 4	137 ± 2	137 ± 4	0.89
Potassium, mmol/L	4.3 ± 0.9	4.0 ± 0.5	4.3 ± 0.9	0.40
Calcium, mmol/L	2.3 ± 0.2	2.4 ± 0.1	2.3 ± 0.2	0.52
Magnesium, mmol/L	0.88 ± 0.13	0.90 ± 0.06	0.88 ± 0.14	0.81
NT-proBNP, pg/mL	3726 [2104–6704]	3510 [1944–5589]	3926 [2101–7258]	0.82
Albumin, g/L	41 ± 7	43 ± 6	41 ± 7	0.41
ASAT, U/L	31 [26–44]	27 [23–32]	31 [27–44]	0.11
ALAT, U/L	30 [21–44]	15 [14–34]	30 [21–44]	0.06
Amylase, U/L	54 [43–71]	39 [22–53]	55 [44–75]	0.054
Lipase, U/L	37 [25–64]	23 [14–25]	39 [26–71]	0.003
FE-1, µg/g	393 ± 98	171 ± 35	413 ± 73	<0.001

Legend: Data are presented as mean ± standard deviation, median [interquartile range], or count (%). Abbreviations: EPI: exocrine pancreatic insufficiency; BMI: body mass index; HF: heart failure; NYHA: New York Heart Association; LVEF: left ventricular ejection fraction; LVEDD: left ventricular end-diastolic diameter; TAPSE: tricuspid annular plane systolic excursion; RV S’: right ventricle systolic excursion velocity; CRP: C-reactive protein; eGFR: estimated glomerular filtration rate; NT-proBNP: N-terminal pro-B-type natriuretic peptide; ASAT: aspartate aminotransferase; ALAT: alanine aminotransferase; FE-1: fecal elastase 1.

**Table 2 nutrients-17-00056-t002:** Micronutrients in patients with heart failure and exocrine pancreatic insufficiency.

	1	2	3	4	5
Vitamin A, µmol/L	1.8	1.3	2.7	1.8	1.1 ^−^
Vitamin D3, nmol/L	62.2	40.1 ^−^	101.7	57.7	62.1
Vitamin E, µmol/L	41.0 ^+^	20.2	27.9	35.8 ^+^	21.3
Selenium, µg/L	42.2 ^−^	71.4	59.9 ^−^	73.3	65.3 ^−^
Zinc, µmol/L	7.3 ^−^	12.4	11.8	13.3	13.2

Legend: +: value above upper limit; −: value below lower limit.

## Data Availability

The data presented in this study are available on request from the corresponding author. The data are not publicly available due to privacy considerations.

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
