# Peer review of "Micronutrient Deficiencies in Heart Failure and Relationship with Exocrine Pancreatic Insufficiency"

_nutrients, 2024, doi:10.3390/nu17010056_

Round 1

Reviewer 1 Report

Comments and Suggestions for Authors

Thank you for the opportunity to review your paper.

Initially I struggled with the rationale for the study. I can not find robust data on how the pancreas is related to selenium and zinc absorption in the GI tract, it is certainly not well understood or described. Selenium has no GI mechanisms to control absorption for example. Therefore to begin with the rationale confused me, and I could not find other sources to confirm this was a good idea, and the actual plan to do it also was difficult to understand how that would answer what question. No national guidelines (looking at UK for example) require zinc or selenium monitoring in chronic pancreatitis or supplementation.

The lack of mention of power I wonder is also concerning. Whilst completely understanding this was a consecutive patient cohort, and therefore you could not control the cohorts, one would expect a protocol that states what comparisons you may or may not do depending on cohort size. This is absent and I am concerned by the number of comparisons particularly as the EPI group was so small, and significantly sicker (and 20 kg lighter). Should this therefore just be descriptive and remove all analysis and discussion of differences because the groups are too different to compare? Though not sure actually what the rationale for the comparison is still. Is it that people with EPI may be more malnourished than people without (just limiting it to cohorts with co-existing heart failure)? 

The methods were great in the fact you mentioned analytical techniques and performance. However there were lots of results reported that were not in the methods e.g. BNP, LFT, ejection fraction. You state later no one was on vitamin supplements but I missed this in the exclusion criteria or methods or results so remain confused if we knew this for sure and how. 

There also needs to be a significant addition in regard to pre analytics. For example zinc is affected by albumin concentration, diurnal variation, time of sampling after a meal, type of blood tube used (o ring present or not). Most of these markers are affected by the acute phase, and many by albumin, and vit E by cholesterol concentration (and similar to zinc etc). Although CRP is indicative of inflammation I am sure you would agree it is not the only marker, and may lack sensitivity if synthetic function of liver is impaired by heart failure etc. Therefore I think there is a major limitation in the fact that I assume pre-analytical factors were not controlled for and you only have single measurements at ?time.

there are also publications that for selenium for example there should be age related ranges, I.e different if older than 65 and there is no mention of all of these common causes of variation in concentration that would affect the results and were not accounted for.  Why did you not use red cell selenium or tocopherol to cholesterol ratios? I note you quote less that 50 for D deficient. Diagnostic criteria have changed in the UK, and actually you do mention this in your discussion for other analytes, that deficiency is only confirmed if there are also symptoms and these get better on treatment. A low concentration does not mean the person is deficient, the limit has moved to less that 25 nmol/l in the UK now as many many studies show there is no benefit from supplementing people with biochemically low result but no symptoms so using a lower threshold increase slightly the chance that some may actually be deficient. Symptoms/syndromes are still required though.

this also raises one areawhich I thought was most challenging, the recommendations at the end of each segment of your discussion. Personally I would remove them as they seem very positive that there is something here and that is not what you have shown and I think they are at odds with the literature. I do note though that papers are quoted that support your suggestions to a degree but I think they are too selective. Meta-analyses show vit A and E supplementation increase mortality and selenium is neutral. Though I do note you state for selenium it only worked in the symptomatic. There have also been large numbers of trials in this area and as we know in general supplements of micronutrients don’t work/are dangerous (outside of true vitamin/mineral deficiency of course) so I don’t think the conclusion to do more studies is ethical or proven by your data.

It is interesting there was a lot of EPI in your cohort so should low body weight in heart failure patients not be attributed to heart failure without ruling out other causes? Is that an important message instead? Could you say that although you don’t know as the numbers were so small there were no significant biochemical indicators I.e. lft and albumin do not indicate EPI etc?  Could you say that despite apparently having never had any history of pancreatitis therefore unknown duration of a condition associated with malabsorption they actually seemed to be not that deficient nutritionally? Therefore have you potentially discovered a spurious cause of low faecal elastase? Or do you think the pt parameters fits with a true diagnosis of missed EPI? You don’t actually Discuss causes of falsely low or high elastase but in the lab we will quote a low elastase may need repeating but a high is usually reassuring enough to be done once only. Particularly as you have a cohort with no symptoms or risk factors for EPI.

The manuscript was well written in the fact I followed through mostly what you were thinking and doing even though I am not sure the papers you reference support this study and have some concerns as listed above, it comes across well though. It certainly made me think, but as you can see this data made me think there was no point looking at cohort comparison as the groups were too different and I suspect your control of factors that affect micronutrient status were not controlled for properly and we have some of the wrong measures but I did think your work on EPI and whether this is a true diagnosis or not in heart failure was interesting. Were there any repeat elastases? Did the EPI cohort get any imaging etc later to confirm pancreatic loss. Was there any classification later of the cause I.e smoking, autoimmune etc etc? 

Author Response

Summary

First of all, we would like to thank you for your careful review and the positive remarks regarding our study, and for the extended, critical assessment of our manuscript. Below you will find our responses to each of your comments.

Point-by-point response to Comments and Suggestions for Authors

Comment 1: Initially I struggled with the rationale for the study. I can not find robust data on how the pancreas is related to selenium and zinc absorption in the GI tract, it is certainly not well understood or described. Selenium has no GI mechanisms to control absorption for example. Therefore to begin with the rationale confused me, and I could not find other sources to confirm this was a good idea, and the actual plan to do it also was difficult to understand how that would answer what question. No national guidelines (looking at UK for example) require zinc or selenium monitoring in chronic pancreatitis or supplementation.

Response 1: We appreciate your concerns regarding the limited understanding and the difficulty in finding robust data connecting the pancreas to selenium and zinc handling/metabolism. The rationale for including selenium and zinc in our study stems from the hypothesized link between EPI and micronutrient deficiencies, which may contribute to poor clinical outcomes in heart failure patients. Although the mechanisms are not fully understood, prior studies suggest reduced exocrine function affects zinc metabolism [1], absorption [2,3] and increases excretion [2]. Additionally, treatment of exocrine insufficiency by enzyme replacement therapy improves (fractional) absorption of zinc [4]. Similarly, pancreas exocrine dysfunction is known to be a risk factor for selenium deficiency [5,6], Although we agree the exact mechanisms remain unknown, we therefore added the following to the discussion:

‘In patients with EPI in the context of chronic pancreatitis or pancreaticoduodenectomy (Armstrong et al. Pancreatology, 2007), a selenium deficiency was reported in 15% and 56% respectively [40]. Nevertheless, the underlying mechanism - the role of exocrine secretions in (intestinal) selenium handling is not well understood.’ (Line 254-258).

Additionally, UK guidelines do take both selenium and zinc into account (see Table 2 and Table 3 in Consensus for the management of pancreatic exocrine insufficiency: UK practical guidelines – 2021 [7]).

This reference was again added to the introduction for clarification:

‘… focusing on fat-soluble vitamins (A, D, and E), as well as the minerals selenium and zinc, since these micronutrients depend on the exocrine pancreas for absorption [13]. Additionally, …’ (line 55)

Finally, concerning your comment on the study rationale: the primary aim is to evaluate micronutrient status in patients with HF where secondarily the focus is on the association with EPI.

Comment 2: The lack of mention of power I wonder is also concerning. Whilst completely understanding this was a consecutive patient cohort, and therefore you could not control the cohorts, one would expect a protocol that states what comparisons you may or may not do depending on cohort size. This is absent and I am concerned by the number of comparisons particularly as the EPI group was so small, and significantly sicker (and 20 kg lighter). Should this therefore just be descriptive and remove all analysis and discussion of differences because the groups are too different to compare? Though not sure actually what the rationale for the comparison is still. Is it that people with EPI may be more malnourished than people without (just limiting it to cohorts with co-existing heart failure)? 

Response 2: We appreciate your concern about the small sample size and the challenges it presents for statistical comparisons. Indeed, the small EPI group and the significant baseline differences between the groups (weight and disease severity, which could be the result of EPI, see Vijver et al. 2024 [8] , attached as PDF) pose limitations, which we have acknowledged in our discussion (lines 307-308). Nevertheless, while the small sample size limits definitive conclusions, these findings provide preliminary, mostly descriptive, insights into the distribution of various micronutrients in patients with heart failure and, given the importance of the pancreas in intestinal absorption, the association with EPI. Notably, this is one of the largest cohorts to date in which vitamin A and E levels have been assessed.

The current study concerns a secondary analysis of a dataset on the prevalence and characteristics of EPI in HF (see attached PDF-file [8]). Since this is a secondary analysis, our initial power calculation was focused on the primary objective of determining the prevalence of EPI in heart failure, not on subgroup comparisons for micronutrient deficiencies. The initial power calculation was based on expected EPI prevalence, not subgroup analyses. No prior studies investigating the relationship between EPI and micronutrient deficiencies in patients with heart failure have been performed. The prevalence of EPI in older (age> 60) populations without (known) pancreatic disease ranges from 5% to 10% [9,10]. The only prior study that evaluated EPI in HF showed a prevalence of 30-40% in a non-advanced HF population [11]. At an anticipated prevalence of 45% in the advanced HF population, 24 patients per group are needed maintain a power (β) of 80% and an α of 0.05 [12]. To make proper comparisons, we aimed to include 60 patients in total.

To clarify, we added the following to the study population section of the Materials & Methods:

… and NT-proBNP). ‘Because this study concerns a secondary analysis of a study on the prevalence of EPI in HF, no formal sample size calculation was performed.’ (lines 68-70)

Eventually, we found less patients with EPI in our cohort than expected. As you mention, however, patients with EPI were significantly sicker and showed more signs of malnourishment. Due to limited sample of patients with EPI, we are unable to draw definitive conclusions regarding malabsorption subsequent to EPI as a possible aetiology of micronutrient deficiencies, based on the current data; we mention this in the discussion of limitations as follows:

‘Secondly, we identified only five patients with EPI who were significantly more malnourished, limiting the ability to interpret possible associations between nutrient deficiencies.’ (lines 307-308)

Despite this, these data still give us new insights in this underreported scientific field and are sufficient to answer the primary aim: to evaluate micronutrient status in patients with HF.

Comment 3: The methods were great in the fact you mentioned analytical techniques and performance. However there were lots of results reported that were not in the methods e.g. BNP, LFT, ejection fraction. You state later no one was on vitamin supplements but I missed this in the exclusion criteria or methods or results so remain confused if we knew this for sure and how. 

Response 3: Thank you for you compliments on the methods and analytical techniques. We appreciate your observation regarding methodological clarification for reported results and added the following in the Materials & Methods:

‘… of enrolment, were excluded. For these patients medical history, clinical characteristics were retrieved from the patients' medical file, as well as available echocardiographic data and laboratory measurements (including kidney and liver function, and NT-proBNP). All patients provided written …’

Secondly, regarding vitamin supplementation, one patient used multivitamins, and five patients used vitamin D, as indicated in Table 1. We found no significant differences in vitamin status between patients with and without EPI, even when these individuals were excluded from the analysis.

Comment 4: There also needs to be a significant addition in regard to pre analytics. For example zinc is affected by albumin concentration, diurnal variation, time of sampling after a meal, type of blood tube used (o ring present or not). Most of these markers are affected by the acute phase, and many by albumin, and vit E by cholesterol concentration (and similar to zinc etc). Although CRP is indicative of inflammation I am sure you would agree it is not the only marker, and may lack sensitivity if synthetic function of liver is impaired by heart failure etc. Therefore I think there is a major limitation in the fact that I assume pre-analytical factors were not controlled for and you only have single measurements at ?time.

Response 4: We agree with your comments regarding the various factors influencing the micronutrients and fully recognize the impact of these factors on the interpretation of micronutrients levels. Unfortunately not all the requested data is available.

When comparing EPI and non-EPI patients: albumin was not significantly different between the groups. CRP was also not different between the groups, but, we agree, there are indeed more markers for inflammation. For example, data on lymphocyte count was also available in most patients, which was also similar between the groups (1.2 ± 0.6 versus 1.3 ± 0.7, p=0.78). There is no data regarding specific interleukins to measure inflammation. Patients were included in a stable (HF) phase during hospital admission.

To assess (micro)nutrient status and malnutrition, it is a definitely a limitation that the abovementioned variables are not controlled for, however we believe doing so is outside the scope of the current manuscript.

Due to logistical obstacles, no trace-element free tubes were available. All samples were processed within an hour to minimalize the effects of the O-ring present in the EDTA-tubes on zinc levels. Additionally, there was no haemolysis present in our samples. We considered the obtained measurements of sufficient quality to answer our research question.

To clarify we added the following to the Materials & Methods:

‘… selenium and zinc, collected using an EDTA-tube. Samples were processed within 1 hour of collection to prevent haemolysis. Afterwards samples were frozen to -20°C before further assessment. Vitamins A, D and E concentrations were …’ (lines 76-78).

And the following to the limitations:

Lastly, several micronutrients may be influenced by pre-analytical factors, such as acute-phase response, protein binding, and time of sampling, which were not controlled for in this study. This may contribute to variability in the measured concentrations.’ (lines 314-317)

Comment 5: there are also publications that for selenium for example there should be age related ranges, I.e different if older than 65 and there is no mention of all of these common causes of variation in concentration that would affect the results and were not accounted for. Why did you not use red cell selenium or tocopherol to cholesterol ratios?

Response 5: Thank you for your insightful comments regarding age-related ranges for selenium and the use of alternative measurements, such as red cell selenium or tocopherol-to-cholesterol ratios.

We measured selenium according to our lab standard and according to the available literature of selenium in HF, which is to measure selenium in plasma concentrations [13]. Plasma selenium is widely used as a marker in studies of selenium status in HF, and our methodology ensures consistency with prior research.

We acknowledge that selenium levels can vary with age and that age-specific reference ranges could provide additional context for interpreting our data. However, there are limited data on age-specific selenium ranges in HF populations, and incorporating such ranges into this analysis was beyond the scope of the current study.

Comment 6: I note you quote less that 50 for D deficient. Diagnostic criteria have changed in the UK, and actually you do mention this in your discussion for other analytes, that deficiency is only confirmed if there are also symptoms and these get better on treatment. A low concentration does not mean the person is deficient, the limit has moved to less that 25 nmol/l in the UK now as many many studies show there is no benefit from supplementing people with biochemically low result but no symptoms so using a lower threshold increase slightly the chance that some may actually be deficient. Symptoms/syndromes are still required though.

Response 6: Thank you for pointing out the evolving diagnostic criteria for vitamin D deficiency and the differences between national guideline. In the day to day clinic it is complex to diagnose a deficiency and lab values must be correlated to the clinical features of the patient. Since we were primarily interested in the differences between the EPI and the non-EPI group, we felt that a single, absolute cut-off value would be sufficient, regardless of symptoms.

We agree, the criteria for diagnosing vitamin D deficiency are topic of debate. In Dutch guidelines, the current cut-off values for vitamin D are <30 nmol/L for people ≤70 years of age, compared to <50nmol/L for patients >70 years of age [14]. In the UK guidelines, people with a vitamin D level <25nmol/L are considered deficient, and people with a vitamin D level <50nmol/L are considered insufficient. Taken together, we chose to use 50nmol/L as a cut-off point in this study. This also allows us to compare our findings to (older) cardiovascular literature, where this cut-off point is adhered to [15]).

We clarified this in the discussion as follows:

‘More than half of the study population suffered from vitamin D deficiency (<50 nmol/L; according to Dutch guidelines) with none showing elevated levels’ (line 204-205)

Comment 7: this also raises one areawhich I thought was most challenging, the recommendations at the end of each segment of your discussion. Personally I would remove them as they seem very positive that there is something here and that is not what you have shown and I think they are at odds with the literature. I do note though that papers are quoted that support your suggestions to a degree but I think they are too selective. Meta-analyses show vit A and E supplementation increase mortality and selenium is neutral. Though I do note you state for selenium it only worked in the symptomatic. There have also been large numbers of trials in this area and as we know in general supplements of micronutrients don’t work/are dangerous (outside of true vitamin/mineral deficiency of course) so I don’t think the conclusion to do more studies is ethical or proven by your data.

Response 7: We agree our data do not directly substantiate positive recommendations on micronutrients supplementation. We therefore reviewed the following sections of the discussion:

Vitamin A: we did not provide any revisions as the recommendations in our opinion convey the conflicting data: ‘While earlier trials showed that supplementation with vitamin A could reduce cardio-vascular events, more recent research failed to confirm these benefits [27,28]. ……. Further research is needed to clarify the role of vitamin A supplementation in HF patients, particularly in light of the conflicting evidence and the lack of data on serum levels in previous studies’ (line 196 – 202). Since these trials did not measure serum levels of vitamin A, it is not proven that vitamin A supplementation is inferior in patients with a lack of vitamin A. Therefore we suggest that studies which investigate targeted therapy in patients who are vitamin A deficient could still be warranted.

Vitamin D: in our opinion the role of vitamin D remains unclear, since the picture that has emerged still contains conflicting data, as reflected the following: ‘However, despite these correlations and hypothetical properties, recent randomized controlled trials have failed to provide evidence for any beneficial impact on cardiovascular health, including prevention and/or the clinical course of HF [30,34]. This contradiction highlights the need for further research to clarify the role of vitamin D in cardiovascular disease’ (lines 215-219). More studies would not be unethical in our opinion.

Vitamin E: we added the following statement to the discussion:

‘… congestive HF [36]. These mechanisms may account for the absence of vitamin E deficiency despite the presence of EPI, and even explaining the values above the upper limit of normal. The latter could also contribute to the failure of clinical trials to show beneficial effects of  preventive administration of vitamin E on major adverse cardiovascular events [THOPES investigators, NEJM 2000 10.1056/NEJM200001203420302 [16]]. Notably, vitamin levels of the participants were not measured prior, during or after completion of the trials’ (lines 236 – 241).

We do not recommend to execute more studies on vitamin E based on these data.

Selenium: the conflicting results on selenium supplementation in HF are presented. We provided minor changes to convey more research is required.

Zinc: no definite recommendations on supplementation are made. We provide minor changes to convey the evidence is inconclusive and more research is required.

Comment 8: It is interesting there was a lot of EPI in your cohort so should low body weight in heart failure patients not be attributed to heart failure without ruling out other causes? Is that an important message instead? Could you say that although you don’t know as the numbers were so small there were no significant biochemical indicators I.e. lft and albumin do not indicate EPI etc?  Could you say that despite apparently having never had any history of pancreatitis therefore unknown duration of a condition associated with malabsorption they actually seemed to be not that deficient nutritionally? Therefore have you potentially discovered a spurious cause of low faecal elastase? Or do you think the pt parameters fits with a true diagnosis of missed EPI? You don’t actually Discuss causes of falsely low or high elastase but in the lab we will quote a low elastase may need repeating but a high is usually reassuring enough to be done once only. Particularly as you have a cohort with no symptoms or risk factors for EPI.

Response 8: This indeed is a very interesting topic, touching upon the subject of cardiac cachexia. We recently published the manuscript of the original study of this dataset - where we discuss this topic (see attached PDF: Vijver et al, 2024 [8]). We feel it goes beyond the scope of the present study to discuss these possible pathophysiological mechanisms in this paper.

Based on these data we cannot state a causal relation between any of the assessed factors and malabsorption, neither can we state that the patients have EPI due to either HF, or another cause. To exclude as many other causes as possible we excluded patients who were at risk for pancreatic dysfunction due to alcohol, previous inflammatory diseases, pancreatic malignancies and who had a history of gall stones. The observations in this study underscore the fact that EPI may be an underrecognized condition in this patient population and warrant further investigation, specifically given the low body weight.

As you correctly point out FE-1 lacks perfect diagnostic accuracy. Although low FE-1 levels are typically indicative of EPI, various factors, including sample handling, can influence the results. Nonetheless, this diagnostic test shows a proper association with the gold standard, i.e. the direct pancreatic function test [17]. For this exploratory study, we considered it unethical to perform an invasive diagnostic test; FE-1 is the most feasible one for the day to day practice. Given the clinical parameters observed in our cohort, we believe it is equally plausible that these patients represent cases of previously undiagnosed EPI. We hope you understand this rationale.

It would strengthen our findings to repeat the measurement like you suggested, however all patients did exhibit symptoms of EPI. A more detailed description in these patients is  described in the other manuscript [8].

Comment 9: The manuscript was well written in the fact I followed through mostly what you were thinking and doing even though I am not sure the papers you reference support this study and have some concerns as listed above, it comes across well though. It certainly made me think, but as you can see this data made me think there was no point looking at cohort comparison as the groups were too different and I suspect your control of factors that affect micronutrient status were not controlled for properly and we have some of the wrong measures but I did think your work on EPI and whether this is a true diagnosis or not in heart failure was interesting. Were there any repeat elastases? Did the EPI cohort get any imaging etc later to confirm pancreatic loss. Was there any classification later of the cause I.e smoking, autoimmune etc etc? 

Response 9: Thank you for this comment, including the positive remarks but also the pitfalls of this study. We agree that there are more factors influencing nutritional status than accounted for in this study, however this real-world data set still provides novel information in a well-defined cohort. We aim to unravel nutritional status in HF and the relationship between the heart and the pancreas, which appears to be complicated for example in terms of diagnostics. We used faecal elastase 1 (FE- 1) measurement; the most employed test to assess the exocrine pancreatic function [18]. We had no repeated FE-1 measurements, but this would be interesting in a follow-up study, in order to relate this to HF severity and micronutritional status. As you suggest, we are currently conducting further studies to better phenotype EPI patients, for example using pancreatic imaging and invasive hemodynamic measurements.

To address your question about the potential causes of EPI in this cohort, we did not perform a classification based on smoking, autoimmune factors, or other risk factors for pancreatic dysfunction (although many patients who were considered at risk were excluded as mentioned above); we encourage future studies to do so. This study primarily focused on micronutrient status and the association with EPI and we hope that this preliminary work will guide further research into this underexplored area.

REFERENCES

1.        Boosalis, M.G.; Evans, G.W.; McClain, C.J. Impaired Handling of Orally Administered Zinc in Pancreatic Insufficiency. Am J Clin Nutr 1983, 37, 268–271, doi:10.1093/ajcn/37.2.268.

2.        Dutta, S.K.; Procaccino, F.; Aamodt, R. Zinc Metabolism in Patients with Exocrine Pancreatic Insufficiency. J Am Coll Nutr 1998, 17, 556–563, doi:10.1080/07315724.1998.10718803.

3.        Hahn, C.; Evans, G. Absorption of Trace Metals in the Zinc-Deficient Rat. American Journal of Physiology-Legacy Content 1975, 228, 1020–1023, doi:10.1152/ajplegacy.1975.228.4.1020.

4.        Easley, D.; Krebs, N.; Jefferson, M.; Miller, L.; Erskine, J.; Accurso, F.; Hambidge, K.M. Effect of Pancreatic Enzymes on Zinc Absorption in Cystic Fibrosis. Journal of Pediatric Gastroenterology &amp Nutrition 1998, 26, 136–139, doi:10.1097/00005176-199802000-00003.

5.        Vaona, B.; Stanzial, A.M.; Talamini, G.; Bovo, P.; Corrocher, R.; Cavallini, G. Serum Selenium Concentrations in Chronic Pancreatitis and Controls. Digestive and Liver Disease 2005, 37, 522–525, doi:10.1016/j.dld.2005.01.013.

6.        Armstrong, T.; Strommer, L.; Ruiz-Jasbon, F.; Shek, F.W.; Harris, S.F.; Permert, J.; Johnson, C.D. Pancreaticoduodenectomy for Peri-Ampullary Neoplasia Leads to Specific Micronutrient Deficiencies. Pancreatology 2007, 7, 37–44, doi:10.1159/000101876.

7.        Phillips, M.E.; Hopper, A.D.; Leeds, J.S.; Roberts, K.J.; McGeeney, L.; Duggan, S.N.; Kumar, R. Consensus for the Management of Pancreatic Exocrine Insufficiency: UK Practical Guidelines. BMJ Open Gastroenterol 2021, 8, e000643, doi:10.1136/BMJGAST-2021-000643.

8.        Vijver, M.A.T.; Dams, O.C.; Gorter, T.M.; van Veldhuisen, C.L.; Verdonk, R.C.; van Veldhuisen, D.J. Exocrine Pancreatic Insufficiency in Heart Failure: Clinical Features and Association with Cardiac Cachexia. Journal of Cachexia, Sarcopenia and Muscle Communications 2024, 7.

9.        Löhr, J.M.; Panic, N.; Vujasinovic, M.; Verbeke, C.S. The Ageing Pancreas: A Systematic Review of the Evidence and Analysis of the Consequences. J Intern Med 2018, 283, 446–460.

10.       Raphael, K.L.; Chawla, S.; Kim, S.; Keith, C.G.; Propp, D.R.; Chen, Z.N.; Woods, K.E.; Keilin, S.A.; Cai, Q.; Willingham, F.F. Pancreatic Insufficiency Secondary to Tobacco Exposure: A Controlled Cross-Sectional Evaluation. Pancreas 2017, 46, 237–243, doi:10.1097/MPA.0000000000000721.

11.       Özcan, M.; Öztürk, Z.; Emet, S.; Aydın, Ş.; Arslan, K.; Arman, Y.; Akkaya, V.; Tükek, T. Evaluation of Malnutrition with Blood Ghrelin and Fecal Elastase Levels in Acute Decompensated Heart Failure Patients. Turk Kardiyol Dern Ars 2015, 43, 131–137, doi:10.5543/tkda.2015.06606.

12.       Viechtbauer, W.; Smits, L.; Kotz, D.; Budé, L.; Spigt, M.; Serroyen, J.; Crutzen, R. A Simple Formula for the Calculation of Sample Size in Pilot Studies. J Clin Epidemiol 2015, 68, 1375–1379, doi:10.1016/j.jclinepi.2015.04.014.

13.       F. Combs, Jr., G. Biomarkers of Selenium Status. Nutrients 2015, 7, 2209–2236, doi:10.3390/nu7042209.

14.       Lips, P.; van Schoor, N.; de Jongh, R. Vitamine D, de Huidige Inzichten. Ned Tijdschr Geneeskd 2020, D3949, 164.

15.       Norman, P.E.; Powell, J.T. Vitamin D and Cardiovascular Disease. Circ Res 2014, 114, 379–393, doi:10.1161/CIRCRESAHA.113.301241.

16.       Investigators, T.H.O.P.E.S. Vitamin E Supplementation and Cardiovascular Events in High-Risk Patients. https://doi.org/10.1056/NEJM200001203420302 2000, 342, 154–160, doi:10.1056/NEJM200001203420302.

17.       Vanga, R.R.; Tansel, A.; Sidiq, S.; El-Serag, H.B.; Othman, M.O. Diagnostic Performance of Measurement of Fecal Elastase-1 in Detection of Exocrine Pancreatic Insufficiency: Systematic Review and Meta-Analysis. Clin Gastroenterol Hepatol 2018, 16, 1220-1228.e4, doi:10.1016/J.CGH.2018.01.027.

18.       Dominici, R.; Franzini, C. Fecal Elastase-1 as a Test for Pancreatic Function: A Review. Clin Chem Lab Med 2002, 40, doi:10.1515/CCLM.2002.051.

Reviewer 2 Report

Comments and Suggestions for Authors

This is a highly interesting study with very important findings. I recommend publishing them after minor changes. I hope my suggestions improve the quality of the paper.

In the introduction, a more comprehensive background on specific nutrient deficiencies in HF should be mentioned.
Why the Authors did not include the supplementation of specific vitamins/mineralas  as a exclusion criteria? Intake of themhem may have a significant effect on  the obtained results and affect bias.
The Micronutrients assessment section does not include information about the blood fraction that was assessed. (as same as point 2.3)
In section 2.4. information about p-value regarding as statistically significant should be added.
How did the authors calculate the sample size?
The mean/median value of nutrientblood concentration is important information. However, to more specific information, the percentage and number of patients with deficiencies and without deficiencies across groups in the tables will be desirable and interesting. Could the authors add the table?
In the Conclusion, some practice information/guidelines could be added.

Author Response

Response to Reviewer 2 Comments

Summary

This is a highly interesting study with very important findings. I recommend publishing them after minor changes. I hope my suggestions improve the quality of the paper.

Thank you for careful review and your appreciation, we agree that these findings are meaningful. We looked into your comments; please find our responses below.

Point-by-point response to Comments and Suggestions for Authors

Comment 1: In the introduction, a more comprehensive background on specific nutrient deficiencies in HF should be mentioned.

Response 1: We agree that background on micronutrient deficiencies is important; therefore this is incorporated in the discussion. In the introduction. we added the following to elaborate on micronutrient deficiencies in HF:

‘data on micronutrient status in patients with HF are limited and mostly derived from small scale studies [4], showing deficiencies that are related to disease severity’ (lines 35-36), 

and:  

‘Proposed mechanisms of micronutrient deficiencies in HF are poor dietary habits, hormonal and/or catabolic imbalance, increased excretion and malabsorption [3,5].’ (line 39).

We have chosen to focus on possible mechanisms underlying micronutrient deficiencies. We dive into the specific nutrient deficiencies in the discussion, to directly relate our findings to the literature. We hope you understand this choice.

Comment 2: Why the Authors did not include the supplementation of specific vitamins/mineralas  as a exclusion criteria? Intake of themhem may have a significant effect on  the obtained results and affect bias.

Response 2: In this study we focused on malabsorption as a mechanism of poor micronutritional status. It can be reasoned that despite appropriate intake, micronutrient levels could still be inadequate. Therefore, we did not exclude patients who took additional vitamins.

We reported – the small amount of – patients who took additional supplements (multivitamins: n=1, vitamin D: n=5). We found similar results when we excluded these patients, i.e. no significant differences in vitamin status between patients with and without exocrine pancreatic insufficiency. However, in future studies evaluating the mechanisms of malabsorption and/or deficiencies, is would be valuable to take total dietary intake into account. See also our statement in the limitations: ‘Fourthly, dietary intake was not analysed, which could have provided more insight in the underlying aetiologies of nutrient deficiencies’. (312)

Comment 3: The Micronutrients assessment section does not include information about the blood fraction that was assessed. (as same as point 2.3)

Response 3: Micronutrients were assessed in plasma samples, not serum. Thank you for this sharp comment. This has been amended throughout the manuscript.

Comment 4: In section 2.4. information about p-value regarding as statistically significant should be added.

Response 4: this is already stated in line 108-109, end of paragraph 2.4: ‘A two-sided p-value of less than 0.05 was considered significant in all analyses’.

Comment 5: How did the authors calculate the sample size?

Response 5: The current study concerns a secondary analysis of a study on the prevalence and characteristics of EPI in HF (Vijver et al. 2024, see PDF). Hence, we calculated our sample size based on the expected prevalence of exocrine pancreatic insufficiency. No prior studies investigating the relationship between EPI and micronutrient deficiencies in patients with heart failure have been performed. The prevalence of EPI in older (age> 60) populations without (known) pancreatic disease ranges from 5% to 10% [1,2]. The only prior study that evaluated EPI in HF showed a prevalence of 30-40% in a non-advanced HF population. At an anticipated prevalence of 45% in the advanced HF population, 24 patients per group are needed maintain a power (β) of 80% and an α of 0.05 [3]. To make proper comparisons, we aimed to include 60 patients in total.

To clarify we added the following to the Materials & Methods:

‘Because this study concerns a secondary analysis of a study on the prevalence of EPI in HF, no formal sample size calculation was performed [8].’

Comment 6: The mean/median value of nutrientblood concentration is important information. However, to more specific information, the percentage and number of patients with deficiencies and without deficiencies across groups in the tables will be desirable and interesting. Could the authors add the table?

Response 6: This has been visualized in Figure 1. Upon your request, we also added this to Table 1.

Laboratory

Vitamin A, µmol/L

1.9 [1.4-2.6]

1.8 [1.2-2.3]

2.0 [1.4-2.6]

0.35

Vitamin A deficient, no.

8 (14)

1 (20)

7 (13)

1.00

Vitamin D3, nmol/L

47.2 [26.7-68.6]

62.1 [48.9-82.0]

45.2 [24.2-68.7]

0.23

Vitamin D3 deficient, no.

31 (53)

1 (20)

30 (58)

0.17

Vitamin E, µmol/L

31.5 ± 9.4

29.2 ± 9.1

31.7 ± 9.5

0.58

Vitamin E deficient, no.

0 (0)

0 (0)

0 (0)

-

Selenium, µg/L

68.9 ± 14.0

62.4 ± 12.4

69.5 ± 14.0

0.28

Selenium deficient, no.

36 (61)

3 (60)

30 (58)

1.00

Zinc, µmol/L

12.6 ± 2.8

11.6 ± 2.5

12.7 ± 2.8

0.42

Zinc deficient, no.

7 (12)

1 (20)

6 (86)

1.00

CRP, mg/L

10 [4-20]

8 [5-77]

10 [4-20]

0.93

Comment 7: In the Conclusion, some practice information/guidelines could be added.

Response 7: The main goal of this paper is to raise awareness to the prevalence of various micronutritional deficiencies, and to encourage fellow researchers to study this important field. Due to the limited number of patients studies in this cohort, no new practical guidelines are given, despite the ones already published.  

REFERENCES

1.        Löhr, J.M.; Panic, N.; Vujasinovic, M.; Verbeke, C.S. The Ageing Pancreas: A Systematic Review of the Evidence and Analysis of the Consequences. J Intern Med 2018, 283, 446–460.

2.        Raphael, K.L.; Chawla, S.; Kim, S.; Keith, C.G.; Propp, D.R.; Chen, Z.N.; Woods, K.E.; Keilin, S.A.; Cai, Q.; Willingham, F.F. Pancreatic Insufficiency Secondary to Tobacco Exposure: A Controlled Cross-Sectional Evaluation. Pancreas 2017, 46, 237–243, doi:10.1097/MPA.0000000000000721.

3.        Viechtbauer, W.; Smits, L.; Kotz, D.; Budé, L.; Spigt, M.; Serroyen, J.; Crutzen, R. A Simple Formula for the Calculation of Sample Size in Pilot Studies. J Clin Epidemiol 2015, 68, 1375–1379, doi:10.1016/j.jclinepi.2015.04.014.

Reviewer 3 Report

Comments and Suggestions for Authors

After careful consideration, I fell that the manuscript entitled Micronutrient Deficiencies in Heart Failure and Relationship with Exocrine Pancreatic Insufficiencyis not suitable for publication as it currently stands. The main point is under the statistical analysis. The linear regression analysis is inadequate and the logistic regression analysis is redundant. Furthermore, the analysis in Table 1 does not seem to be correct and should be revised. Consequently, every manuscript should be revised in case the change in the results of the analyses alters any of the conclusions/discussions in the paper.

Here are my comments:

1. Statistical analysis section: It was not reported which test (KS, Shapiro-wilk, etc.) was adopted to verify the normality of the data.

2. Legend of Figure 1. The authors stated that “Distribution of nutrients are indicated by the upper blue boxplot, displaying the... means with standard deviations for vitamin E, selenium and zinc.” But this statement doesn't seem right to me, since the boxplots are not symmetrical.  In fact, I believe that these boxplots are representing the median and interquartile ranges, as described in the figure's (inside) legend.

3. Lines 140-142: Since only a simple regression analysis was performed and some of the response variables do not follow a normal distribution (vitamin A, for example), it would be more correct to calculate the correlation between the variables. Pearson's correlation for normal responses and Spearman's correlation for non-normal responses. In fact, in the case of normality, the result of Pearson's correlation is mathematically equivalent to simple linear regression analysis. Also note that as Sparman's correlation is based on ranks, the correlation value of the raw values or with logarithmic transformation is exactly the same, making the analysis coherent. The same applies to the supplementary table.

4. Table 2: The logistic regression in Table 2 is unnecessary (and can be excluded from the paper), as it is similar to the analysis in Table 1. All conclusions can be derived from Table 1. In fact, the results in Table 1 should be consistent with those in Table 2 (see comment below).

5. Table 1. The results in Table 1 should be reviewed, as I believe they are not correct. Take for example the variable Vitamin E (mean=29.2 ; sd= 9.1; n=5) vs (mean =31.7 ; sd= 9.5; n=54). In this case, p=0.39 (considering the non paired T-test) is incorrect. In fact, the correct value is 0.60 (the results are only not equal because in logistic regression the statistic test is approximated by the normal, and in the T-test the Student's t-distribution is adopted. However, in essence the two tests have the same objective. In addition, due to the small number of observations in one of the groups, n=5, the results of the logistic regression may be inaccurate, which makes it better to use the t-test). Therefore, the entire analysis in Table 1 should be revised and the results corrected, including throughout the text. Consider that this correction may alter some of the conclusions/discussions in the results. In addition, the statistical analysis section should be reformulated after excluding the regression analyses and including the correlation calculations.

Author Response

Response to Reviewer 3 Comments

Summary

After careful consideration, I fell that the manuscript entitled “Micronutrient Deficiencies in Heart Failure and Relationship with Exocrine Pancreatic Insufficiency” is not suitable for publication as it currently stands. The main point is under the statistical analysis. The linear regression analysis is inadequate and the logistic regression analysis is redundant. Furthermore, the analysis in Table 1 does not seem to be correct and should be revised. Consequently, every manuscript should be revised in case the change in the results of the analyses alters any of the conclusions/discussions in the paper.

Thank you for taking the time to review this manuscript, especially for your critical and thorough assessment of the statistical methods. Please find the detailed responses below and in the tracked changes in the re-submitted files. Changes made in the text of the manuscript are stated in red, bold and underlined.

Point-by-point response to Comments and Suggestions for Authors

Comment 1. Statistical analysis section: It was not reported which test (KS, Shapiro-wilk, etc.) was adopted to verify the normality of the data.

Response 1: normality was checked by visual assessment of Q-Q plots. The following has been added to the methods of the manuscript:

‘2.4. Statistical analyses

Normality of the data was checked by visual assessment of Q-Q plots. Data that adhere to a normal distribution …’ (line 97)

Comment 2. Legend of Figure 1. The authors stated that “Distribution of nutrients are indicated by the upper blue boxplot, displaying the... means with standard deviations for vitamin E, selenium and zinc.” But this statement doesn't seem right to me, since the boxplots are not symmetrical.  In fact, I believe that these boxplots are representing the median and interquartile ranges, as described in the figure's (inside) legend.

Response 2: The distributions of vitamin E, selenium and zinc are normal. The asymmetry that arises is because of the black lines that represent the minimum and maximum (as stated in the legend) but despite these outliers the overall distribution is normal and the mean (= also median) lies in the centre of the blue box. However, we understand the confusion and therefore added another colour to distinguish the mean and the standard deviation for the normally distributed values from the medians and IQR’s.

Figure 1. Visual representation of distribution of nutrients in patients with HF compared to population-based reference values.

Comment 3. Lines 140-142: Since only a simple regression analysis was performed and some of the response variables do not follow a normal distribution (vitamin A, for example), it would be more correct to calculate the correlation between the variables. Pearson's correlation for normal responses and Spearman's correlation for non-normal responses. In fact, in the case of normality, the result of Pearson's correlation is mathematically equivalent to simple linear regression analysis. Also note that as Sparman's correlation is based on ranks, the correlation value of the raw values or with logarithmic transformation is exactly the same, making the analysis coherent. The same applies to the supplementary table.

Response 3: thank you for reviewing this for accuracy. We agree with the reviewer that we have to alter the analysis to take the distribution of the variables into account. We already log transformed the predictors in the simple regression analysis (e.g. duration of heart failure, NT-proBNP). We failed to take this into account for the dependent variables (vitamin A and D), therefore we applied the log transformation to the univariate regression model. Because of the cases where the original value was between 0 and 1 (n=3 for vitamin A), the entire variable was shifted (log(vitamin A + 1)). See the revised tables below with the altered description underneath, also updated in the supplementary material:

Supplementary Table S1: Linear regression vitamin A, D, E, zinc and selenium

S1.1 Univariate linear regression vitamin A (retinol)**

B

S.E.

β

95% CI

p-value

Age

0.002

0.01

0.19

-0.001, 0.004

0.16

Weight

0.000

0.001

0.06

-0.002, 0.003

0.67

BMI

0.001

0.004

0.04

-0.007, 0.009

0.76

Duration of HF*

0.01

0.29

-0.03

-0.052, 0.064

0.83

NT-proBNP*

0.19

0.04

0.07

-0.051, 0.088

0.59

FE-1

0.000

0.000

0.08

0.000, 0.000

0.55

Amylase*

0.09

0.07

0.17

-0.050, 0.238

0.20

Lipase*

0.05

0.07

0.10

-0.086, 0.182

0.77

LVEF

-0.001

0.001

-0.06

-0.003, 0.002

0.65

Cardiac index*

-0.06

0.18

-0.06

-0.427, 0.306

0.74

S1.2 Univariate linear regression vitamin D (D3)*

Age

0.01

0.003

0.21

-0.001, 0.010

0.13

Weight

-0.01

0.002

-0.29

-0.010, -0.001

0.03

BMI

-0.02

0.01

-0.37

-0.040, -0.007

0.01

Duration of HF*

0.6

0.06

0.12

-0.069, 0.187

0.34

NT-proBNP*

0.04

0.08

0.07

-0.117, 0.192

0.63

FE-1

0.00

0.00

0.08

-0.001, 0.001

0.57

Amylase*

0.19

0.16

0.16

-0.231, 0.516

0.23

Lipase*

0.01

0.15

0.01

-0.295, 0.307

0.97

LVEF

0.11

0.06

0.23

-0.015, 0.235

0.08

Cardiac index*

0.59

0.44

0.23

-0.97, 1.483

0.18

Legen table S1: β: regression coefficient, S.E.: standard error, OR: odds ratio, CI: confidence interval. *: logarithmic transformation, ** logarithmic transformation + 1.

The following was added to the statistical analyses section:

‘… into a logarithmic scale. To variables containing values between 0 and 1, a constant of 1 was added to ensure all transformed values were suitable for transformation. A two-sided … ‘ (line 106-107)

The results depicted in the revised table have also been updated in manuscript:

‘Vitamin D levels were negatively associated with weight and BMI (p=0.03 and p=0.01, respectively), whilst vitamin E levels were positively associated with age.’ (line 153-154)

Comment 4. Table 2: The logistic regression in Table 2 is unnecessary (and can be excluded from the paper), as it is similar to the analysis in Table 1. All conclusions can be derived from Table 1. In fact, the results in Table 1 should be consistent with those in Table 2 (see comment below).

Response 4: see Response 5.

Comment 5. Table 1. The results in Table 1 should be reviewed, as I believe they are not correct. Take for example the variable Vitamin E (mean=29.2 ; sd= 9.1; n=5) vs (mean =31.7 ; sd= 9.5; n=54). In this case, p=0.39 (considering the non paired T-test) is incorrect. In fact, the correct value is 0.60 (the results are only not equal because in logistic regression the statistic test is approximated by the normal, and in the T-test the Student's t-distribution is adopted. However, in essence the two tests have the same objective. In addition, due to the small number of observations in one of the groups, n=5, the results of the logistic regression may be inaccurate, which makes it better to use the t-test). Therefore, the entire analysis in Table 1 should be revised and the results corrected, including throughout the text. Consider that this correction may alter some of the conclusions/discussions in the results. In addition, the statistical analysis section should be reformulated after excluding the regression analyses and including the correlation calculations.

Response 5: Thank you for pointing this out. The appropriate changes have been made and alle the variables have been reviewed. This led to changes in vitamin E, selenium and zinc (see tracked changes in Table 1). The sections referring to these outcomes have been changed in the results section as well (i.e. removed ‘However, patients with EPI exhibited a trend towards lower selenium levels (62.4 ± 12.4 µg/L versus 69.5 ± 14 µg/L, p=0.17).’ Furthermore, we agree that the added value of the logistic regression (Table 2) is limited, so we agreed to eliminate this table from our manuscript and removed the following sentence: ‘Logistic regression analysis did not show a significant association between micronutrient status and EPI (Table 2).’

Round 2

Reviewer 3 Report

Comments and Suggestions for Authors

The authors made the corrections and/or justified their comments satisfactorily. Therefore, I believe that the manuscript is now suitable for publication.